# Imaging-based indices combining disease severity and time from disease onset to predict COVID-19 mortality: A cohort study

**Giulia Besutti**[1]*, **Olivera Djuric**[2], **Marta Ottone**[2], **Filippo Monelli**[1,3], **Patrizia Lazzari**[4], **Francesco Ascari**[4], **Guido Ligabue**[4], **Giovanni Guaraldi**[5], **Giuseppe Pezzuto**[6], **Petra Bechtold**[7], **Marco Massari**[8], **Ivana Lattuada**[9], **Francesco Luppi**[9], **Maria Giulia Galli**[9], **Pierpaolo Pattacini**[1], **Paolo Giorgi Rossi**[2]

1 Radiology Department, AUSL—IRCCS di Reggio Emilia, Reggio Emilia, Italy, 2 Epidemiology Unit, AUSL–IRCCS di Reggio Emilia, Reggio Emilia, Italy, 3 Clinical and Experimental Medicine University of Modena and Reggio Emilia, Modena, Italy, 4 Department of Radiology, AOU Policlinico di Modena, University of Modena and Reggio Emilia, Modena, Italy, 5 Department of Infectious Diseases, AOU Policlinico di Modena, University of Modena and Reggio Emilia, Modena, Italy, 6 Emergency Department, AOU Policlinico di Modena, Modena, Italy, 7 Epidemiology and Risk Communication Unit, Department of Public Health, Local Health Unit, Modena, Italy, 8 Infectious Disease Unit, Arcispedale Santa Maria Nuova, Azienda USL-IRCCS di Reggio Emilia, Reggio Emilia, Italy, 9 Emergency Department, AUSL-IRCCS di Reggio Emilia, Reggio Emilia, Italy

* giulia.besutti@ausl.re.it

**Data Availability Statement:** According to Italian law, anonymized data can only be made publicly available if there is potential for the re-identification of individuals (https://www.garanteprivacy.it).

## Abstract

### Background

COVID-19 prognostic factors include age, sex, comorbidities, laboratory and imaging findings, and time from symptom onset to seeking care.

### Purpose

The study aim was to evaluate indices combining disease severity measures and time from disease onset to predict mortality of COVID-19 patients admitted to the emergency department (ED).

### Materials and methods

All consecutive COVID-19 patients who underwent both computed tomography (CT) and chest X-ray (CXR) at ED presentation between 27/02/2020 and 13/03/2020 were included. CT visual score of disease extension and CXR Radiographic Assessment of Lung Edema (RALE) score were collected. The CT- and CXR-based scores, C-reactive protein (CRP), and oxygen saturation levels ($sO_2$) were separately combined with time from symptom onset to ED presentation to obtain severity/time indices. Multivariable regression age- and sex-adjusted models without and with severity/time indices were compared. For CXR-RALE, the models were tested in a validation cohort.

### Results

Of the 308 included patients, 55 (17.9%) died. In multivariable logistic age- and sex-adjusted models for death at 30 days, severity/time indices showed good discrimination ability, higher

Furthermore, property of the data remains of the patient, who gave consent to use data for the objective of the study. Thus, data cannot be shared publicly. However, the data underlying this study are available on request to researchers who meet the criteria for access to confidential data (even if anonymous data are provided, they should be published in aggregated form) and for studies with objectives consistent with those of the original study. In order to obtain data, approval must be obtained from the Area Vasta Emilia Nord (AVEN) Ethics Committee, who would check the consistency of the objective and planned analyses and would then authorize us to provide aggregated or anonymized data. Data access requests should be addressed to the Ethics Committee at CEReggioemilia@ausl.re.it as well as to the authors at the Epidemiology unit of AUSL - IRCCS of Reggio Emilia at info.epi@ausl.re.it, who are the data guardians.

**Funding:** The author(s) received no specific funding for this work.

**Competing interests:** The authors have declared that no competing interests exist.

for imaging than for laboratory measures ($AUC_{CT} = 0.92$, $AUC_{CXR} = 0.90$, $AUC_{CRP} = 0.88$, $AUC_{sO2} = 0.88$). $AUC_{CXR}$ was lower in the validation cohort (0.79). The models including severity/time indices performed slightly better than models including measures of disease severity not combined with time and those including the Charlson Comorbidity Index, except for CRP-based models.

## Conclusion

Time from symptom onset to ED admission is a strong prognostic factor and provides added value to the interpretation of imaging and laboratory findings at ED presentation.

## Introduction

After more than two years since the first case of COVID-19 was detected in December 2019 in China, more than 450 million cases and 6 million deaths have been reported globally up to March 2022 [1]. Although the critical care bed occupancy crisis is expected to subside as immunisation accelerates, because COVID-19 detection and related hospital admissions periodically increase, the pandemic continues to be a challenge.

Much effort has been put into identifying the factors associated with the need for critical care and death, starting from a set of information available at emergency department (ED) admission, including clinical, laboratory, and radiological findings [2, 3].

Among the clinical factors, the most used to predict COVID-19 severity and mortality are older age, pre-existing comorbidities, hypoxia, and laboratory tests indicative of increased inflammatory response, COVID-19-related coagulopathy, and end-organ dysfunction [3–7].

Thoracic imaging with chest radiography (CXR) and computed tomography (CT) is a key tool in the evaluation of the degree of lung involvement. CT is highly sensitive in the diagnosis of COVID-19 pneumonia [8, 9], and CT percentage of lung involvement and the quantitative burden of consolidation are among the most important prognostic factors for short-term prognosis in COVID-19 patients [10–16].

Although CXR is less sensitive than CT in diagnosing COVID-19 pneumonia, the prognostic role of CXR, i.e., its utility in predicting outcomes in patients already diagnosed with COVID-19 pneumonia, is only beginning to be explored. In fact, CXR assessment using either the Brixia score or the Radiographic Assessment of Lung Edema (RALE) score appears to be reliable in predicting the various outcomes of COVID-19 in patients admitted to ED [17–23].

Patients with a rapid worsening of symptoms and a short time between symptom onset and seeking care are more likely to have the worst outcomes [24–26]. However, the combined effect of baseline clinical-laboratory findings and time from symptom onset on COVID-19 patients' outcomes has never been explored. Similarly, CT or CXR-based scoring systems are mostly based on the combined effects of the extent of pulmonary involvement and respective attenuation patterns (i.e., normal, ground-glass opacities—GGO, and consolidation), but they do not consider the time from symptom onset as an important adjunctive factor in outcome prediction.

The aim of this study was to evaluate the role of indices that combine radiological data at admission and time from disease onset, in predicting mortality of COVID-19 patients admitted to the ED.

## Materials and methods

### Setting

The province of Reggio Emilia, located in Northern Italy, has a population of 531,751 inhabitants. There are six hospitals in this province, with an emergency department (ED) in the main hospital in the city of Reggio Emilia as well as in four of the five smaller health district hospitals. There is no private ED service. The first case of SARS-COV-2 in the province was diagnosed on 27 February 2020. The cumulative incidence in the first wave of the pandemic reached 0.9% (March–April 2020). During the first pandemic wave in Reggio Emilia, patients presenting to the ED with fever and SpO2>95% were discharged home in case of negative chest X-rays and/or CT or in case of positive chest X-rays and/or CT scan but who were >70 years of age and had no relevant past medical history. Patients >70 years of age and/or with relevant past medical history could be hospitalized even without respiratory failure, while others were admitted to hospital in case of radiological findings of pneumonia combined with respiratory failure. In the case of radiological findings of complicated pneumonia with or without acute respiratory distress syndrome (ARDS), patients started non-invasive ventilation in the ED and were admitted to subintensive/intensive care unit [27].

### Study design and selection of participants

This retrospective cohort study included all consecutive patients meeting the following inclusion criteria: patients > 18 years of age; presenting to any one of the provincial EDs between 27 February and 13 March 2020; and with a positive RT-PCR within 10 days from ED admission. Patients who did not undergo both CT scan and CXR at ED presentation were excluded. During the COVID-19 outbreak, the diagnostic protocol for suspected COVID-19 patients presenting to EDs included RT-PCR, blood tests, chest X-rays, and CT in every case of suggestive X-rays or negative X-rays but with highly suggestive clinical features.

Even if the researchers assessed inclusion criteria and conditions at admission retrospectively, all the information registered at the time of ED presentation (e.g., laboratory data or CT disease extension) was not modifiable later, therefore exposures could not be influenced by the occurrence of the outcome.

The main outcome was death occurring in the 30 days following ED visit.

Data of the included patients were partially used in previous studies to assess CT diagnostic accuracy and added prognostic value [8, 16].

### Ethical approval

The study was approved by the Ethics Committee Area Vasta Emilia Nord (n.2020/0045199). Given the retrospective nature of the study, the Ethics Committee authorized the use of a patient's data without his/ her informed consent if all reasonable efforts had been made to contact that patient.

### Clinical data

Date of symptom onset, diagnosis, hospitalization, and death were retrieved from the COVID-19 Surveillance Registry, implemented in each Local Health Authority. The surveillance is fed by several sources: the Department of Public Health's epidemiological investigations, contact tracing, and symptom surveillance for people in self-isolation, laboratory reports, electronic ED and hospital records, and death certificates for hospitalized patients. Data from the COVID-19 Surveillance Registry were linked with the provincial Radiology Information

System to search for CT scans performed at the moment of or after the onset of COVID symptoms.

Clinical covariates included patient characteristics (age, sex) and the presence of comorbidities, calculated separately, as well as the Charlson Comorbidity Index (CCI), which provides an overall measure of an individual patient's complexity [28]. We categorized the Index in four classes: 0 (absence of relevant comorbidities), 1, 2, and $\geq$ 3 comorbidities. Information on comorbidities was collected from hospital discharge databases of all hospital admissions occurring in the previous 10 years, before the start of the SARS-CoV-2 pandemic in Italy. Diabetes and cancer diagnoses in people residing in the province of Reggio Emilia were ascertained through linkage with the local Diabetes Registry and the Cancer Registry.

C-reactive protein (CRP) and lactate dehydrogenase (LDH) levels and white blood cell, lymphocyte, neutrophil, and platelet counts, as well as arterial blood gas analysis data, all measured at ER presentation, were collected from the provincial Local Health Authority's Laboratory Information System database. All the tests were carried out in the Reggio Emilia Hospital Clinical Laboratories with routine automated methods.

In order to have a reference for the added value of radiological imaging, we computed the severity/time indices also for two basic laboratory data usually available at ED presentation, i.e., oxygen saturation level ($sO_2$) as a measure of lung damage, and CRP as a measure of the inflammatory process characterizing COVID-19 pneumonia.

## Radiological data

CT scans were performed using one of three scanners (128-slice Somatom Definition Edge, Siemens Healthineers; 64-slice Ingenuity, Philips Healthcare; 16-slice GE Brightspeed, GE Healthcare) without contrast media injection. Scanning parameters were tube voltage 120 KV, automatic tube current modulation, collimation width 0.625 or 1.25 mm, acquisition slice thickness 2.5 mm, and interval 1.25 mm. Images were reconstructed with a high-resolution algorithm at slice thickness 1.0/1.25 mm. The extension of pulmonary lesions estimated by using a visual scoring system, resulting in a percentage of total lung parenchyma which had any pathological changes likely due to COVID-19, was extracted from the verbal and structured CT reports [8].

CXRs were retrospectively reviewed by a single radiologist to collect RALE score [29, 30]. Every CXR was divided into quadrants defined vertically by the vertebral column and horizontally by the first branch of the left main bronchus. An extension score from 0 to 4 (0: none, 1: <25%, 2: 25/50%, 3: 50/75%, 4 >75%) and a density score from 1 to 3 (1: Hazy, 2: Moderate, 3: Dense) were assigned to each quadrant. The RALE score was calculated as the sum of the products of the two scores obtained for each quadrant.

To validate the ability of the CXR-based RALE score to predict death, a validation set from a different province (Modena, Northern Italy) was used: all the consecutive cases presenting at the Policlinico di Modena Hospital ED for suspected pneumonia and with confirmed COVID-19 (i.e., positive for SARS-CoV-2 on RT-PCR) between 24 February and 13 April 2020 for whom a CXR was available. For this set of patients, only age, sex, and time from symptom onset were known.

## Severity/Time indices

In order to obtain indices that could incorporate information on disease severity at ED admission and time from disease onset, we combined the CT visual score, the CXR RALE score, CRP levels, and $sO_2$ levels separately with the time elapsed from symptom onset and the measurement of these parameters. The easiest way to do this was to divide the severity measure by the time needed to reach that level of severity. This strategy was used for CRP levels and for CT- and CXR-based extension of parenchymal involvement, while $sO_2$ severity/time index

was calculated as the difference in $sO_2$ value to 100 (100-$sO_2$) divided by the time from symptom onset to $sO_2$ measurement.

We set a five-day lag time representing the average amount of time during which the disease had progressed before symptom onset, based on the reported time period between SARS-CoV-2 infection and COVID-19 symptom onset [31].

CXR RALE-based severity/time index was also calculated for the Modena validation cohort.

### Statistical analyses

Continuous variables are reported as median and interquartile range, and categorical variables as proportions. Poisson regression models were used to estimate incidence rate ratios (IRR) with 95% confidence intervals (95% CI) for death, unadjusted and adjusted for age and sex.

A multivariable regression model adjusted for age and sex was used to compare models without and with severity/time indices. The comparison of model performances was done by Log likelihood, Akaike's information criteria (AIC), and P value for Z-test for severity/time index. Area under the ROC curve (AUC) was used to estimate predictive value of different severity/time indices. We used Stata 13.0 SE (Stata Corporation, Texas, TX) software package.

Validation was conducted applying the model parameters estimated in the Reggio Emilia cohort for CXR RALE score and respective severity/time index to a cohort of cases collected in the ED of the Policlinico di Modena hospital. AUC is reported to measure the predictivity of the model in a different setting.

## Results

### Characteristics of study subjects

After excluding RT-PCR negative patients or those who did not have a complete radiological assessment within 10 days from ED presentation, 308 patients were included (Fig 1).

Of these 308 COVID-19 patients, 55 (17.9%) died (Table 1); the patients who died were older, generally male, and had a higher prevalence of comorbidities, including cancer, ischemic cardiopathy, hypertension, heart failure, and arrhythmias, with an accordingly higher CCI. Median values of CRP were higher, while $sO_2$ was lower in this group. CT- and CXR-RALE-based extension of pulmonary involvement was higher in those who died. Time from symptom onset to chest imaging was shorter in patients who died. CT visual score and CXR RALE score were strongly associated with Spearman's rho of 0.73 (P < .001) (Fig 2).

### Variables associated with death

Age- and sex-adjusted analyses (Table 2) showed that the variables associated with death were various comorbidities, the strongest associations being for heart failure (IRR 2.68; 95% CI, 1.41–5.10), higher CRP levels (IRR for one mg/dl increase 1.04; 95% CI, 1.01–1.08), lower $sO_2$ levels (IRR for one unit increase 0.97; 95% CI, 0.94–1.00), shorter time from symptom onset to CT or CXR assessment (IRR for one day increase 0.91; 95% CI, 0.83–1.00, and 0.915; 95%CI, 0.84–1.00, respectively), and higher extension of lung involvement assessed both by CT visual score (IRR for one unit increase 1.03; 95% CI, 1.02–1.04) and CXR RALE score (IRR for one unit increase 1.06; 95% CI, 1.03–1.10).

### Indices combining severity and time

Severity/time indices were associated with death after adjusting for age and sex, with IRRs varying from 1.22 to 1.42 (Table 2). As depicted in ROC curves (Fig 3), in multivariable logistic models for death at 30 days adjusted for age and sex, severity/time indices showed good

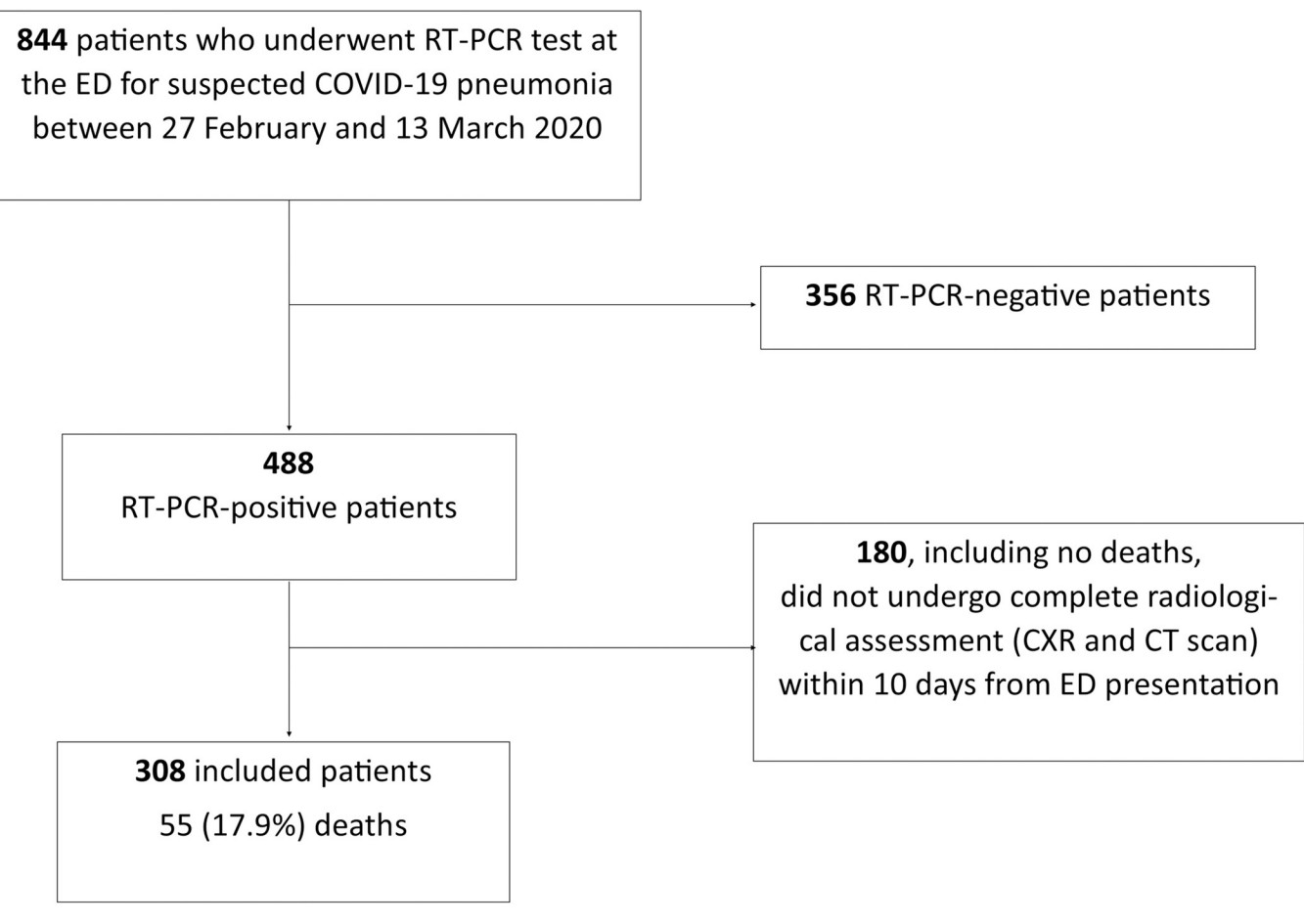

**Fig 1. Flowchart representing patient inclusion.**

discrimination ability, which was higher for radiological than for laboratory measures [$AUC_{CT}$ = 0.92 (95%CI 0.89–0.95), $AUC_{CXR}$ = 0.90 (95%CI 0.86–0.94), $AUC_{CRP}$ = 0.88 (95%CI 0.83–0.93), and $AUC_{sO2}$ = 0.88 (95%CI 0.84–0.92)].

Noticeably, the performance of models including severity/time indices based on the ratio between measures of disease severity and time from symptom onset was slightly better than that of models including the same measures of disease severity not combined with time (Fig 3). For instance, the AUC of the model including CT score only was 0.90 (95% CI 0.87–0.94), increasing to 0.92 (95%CI 0.89–0.95) when substituting CT score with CT-based severity/time index (S1 Table). Moreover, the models including severity/time indices resulted in a minimal information loss when compared to models including the same measures plus time from symptom onset considered separately (S1 Table). Finally, the models including severity/time indices showed a slightly better performance when compared to models including the same measure of disease severity plus the CCI, with the exception of CRP-based models (S1 Table).

## Validation cohort

Age- and sex- adjusted models for death at 30 days including CXR-RALE score alone or combined in a severity/time index with a five-day lag were also evaluated in a cohort of 215 consecutive COVID-19 patients who presented to the ED of an adjacent province (Modena). Of these, 48 (22.3%) died (S2 Table). In this cohort, days from symptom onset were similarly

**Table 1. Patients' pre-existing condition, and clinical, laboratory, and CT findings at admission.**

| Variables | | All Patients | Deaths | |
|---|---|---|---|---|
| | | N (%) | N (%) | P* |
| | | 308 | 55 (17.86) | |
| Age (years), median (IQR) | | 65.4 (52.8–75.7) | 79.7 (72.0–85.0) | <0.001** |
| Female sex | | 119 (38.6) | 13 (23.6) | 0.012 |
| Calendar time | Week 1 | 36 (11.7) | 8 (14.6) | 0.005 |
| | Week 2 | 163 (52.9) | 38 (69.1) | |
| | Week 3 | 109 (35.4) | 9 (16.4) | |
| Charlson Comorbidity Index | 0 | 232 (75.3) | 26 (47.3) | <0.001 |
| | 1 | 21 (6.8) | 6 (10.9) | |
| | 2 | 19 (6.2) | 5 (9.1) | |
| | ≥3 | 36 (11.7) | 18 (32.7) | |
| Diabetes | | 42 (13.6) | 11 (20.0) | 0.129 |
| Cancer | | 49 (15.9) | 14 (25.5) | 0.033 |
| Chronic obstructive pulmonary disease | | 10 (3.3) | 7 (12.7) | <0.001 |
| Ischemic cardiopathy | | 30 (9.7) | 12 (21.8) | 0.001 |
| Chronic kidney failure | | 3 (1.0) | 2 (3.6) | 0.083 |
| Hypertension | | 55 (17.9) | 20 (36.4) | <0.001 |
| Obesity | | 6 (2.0) | 3 (5.5) | 0.072 |
| Heart failure | | 18 (5.8) | 12 (21.8) | <0.001 |
| Arrhythmias | | 23 (7.5) | 11 (20.0) | <0.001 |
| Vascular diseases | | 6 (2.0) | 3 (5.5) | 0.072 |
| CRP (mg/dl), median (IQR) (missing values = 38) | | 5.2 (2.1–11.6) | 11.4 (4.2–15.9) | 0.001 |
| sO$_2$ (%), median (IQR) | | 94.9 (92.8–96.0) | 92.5 (89.6–94.5) | <0.001 |
| Days from symptom onset to CT, median (IQR) | | 7 (4–8) | 5 (2–7) | 0.009** |
| CT disease extension (visual score) | <20% | 107 (34.7) | 7 (12.7) | <0.001 |
| | 20–39% | 110 (35.7) | 12 (21.8) | |
| | 40–59% | 58 (18.8) | 16 (29.1) | |
| | ≥60% | 33 (10.7) | 20 (36.4) | |
| Days from symptom onset CXR, median (IQR) | | 6.5 (4–8) | 5 (2–7) | 0.005** |
| CXR RALE score, median (IQR) | | 9 (5–13.5) | 15 (11–21) | <0.001 |

IQR, interquartile range; CRP, C-reactive protein; sO$_2$, oxygen saturation level; CT, computed tomography; CXR, chest X-ray; RALE, Radiographic Assessment of Lung Edema.

*Pearson's chi-squared test or Fisher exact test and P value for the hypothesis of independence in the two-way table.

** P value nonparametric equality-of-medians test.

associated with death (IRR for one day increase 0.93; 95% CI, .86–1.00), while the association was weaker for RALE score (IRR for one unit increase 1.03; 95% CI, 1.00–1.06) (Table 3). The respective severity/time index in this cohort also showed a higher association (IRR for one unit increase 1.33; 95% CI, 1.01–1.76), and the model including the severity/time index performed slightly better than the model with RALE score (AUC$_{CXR}$ from 0.77 for RALE alone to 0.79 for RALE combined in a severity/time index) (S1 Fig.). However, both these AUC values were noticeably lower than in the Reggio Emilia cohort.

## Discussion

We used a cohort of consecutive COVID-19 patients who presented to the ED and underwent imaging assessment of COVID-19 pneumonia to test various composite indices combining

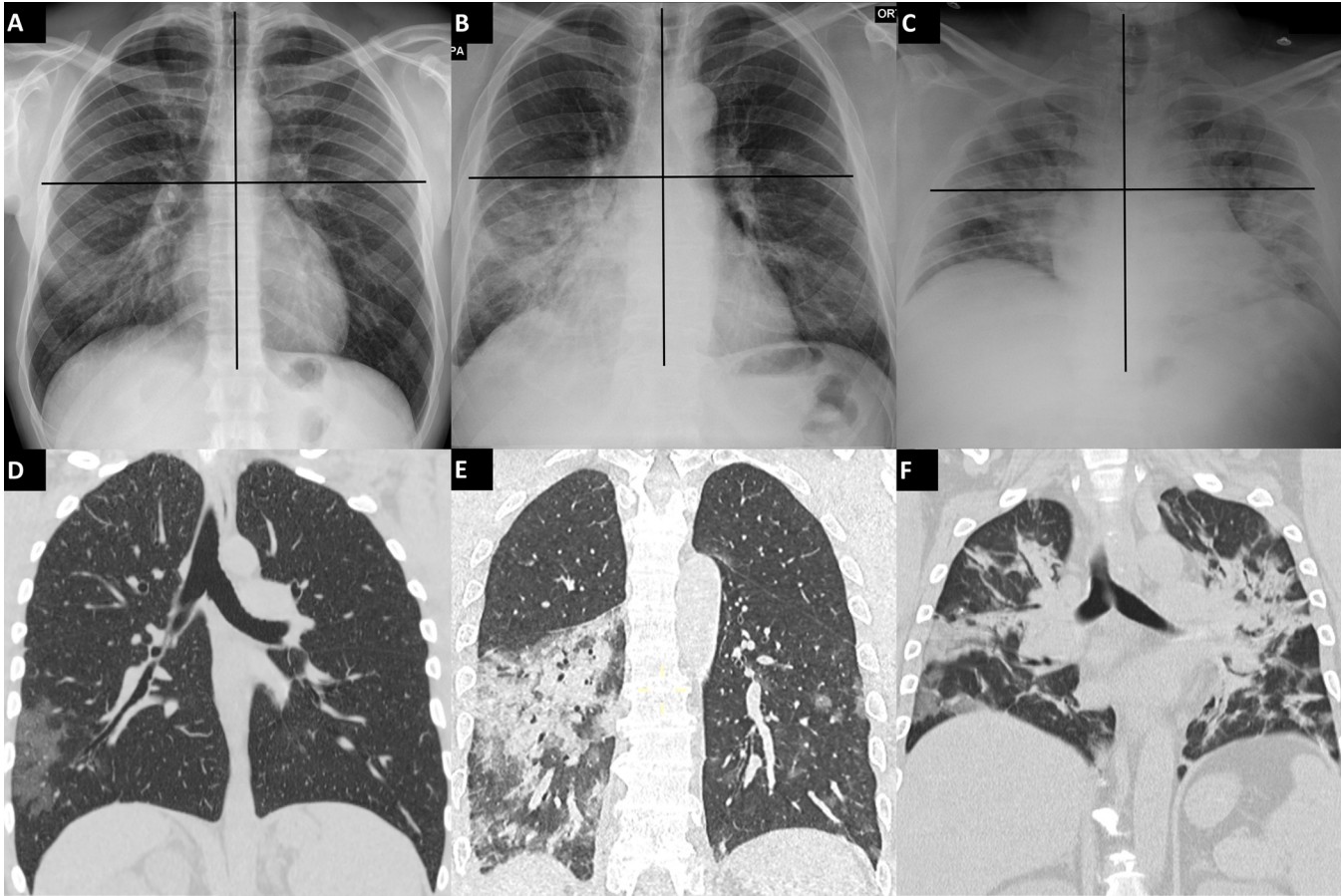

**Fig 2. Example of RALE scores and respective CT.** Examples of Radiographic Assessment of Lung Edema (RALE) on chest X-rays (CXR) which were divided into quadrants (A, B, C) and correlation with coronal reconstruction of chest computed tomography (CT) performed within 24 hours (D, E, F). Patient 1 (male, 37 y/o) was attributed a RALE score of 7 because of the presence of moderate alveolar opacities in 50/75% of the right inferior quadrant and hazy alveolar opacities in < 25% of the left inferior quadrant (A); chest CT demonstrated the presence of GGO in the right lower lobe and a small area of GGO in the left lower lobe (visual score 10%) (D). Patient 2 (male, 53 y/o) had a RALE score of 19 due to dense alveolar opacities in > 75% of the right lower quadrant, moderate alveolar opacities in 50/75% of the left lower quadrant, and hazy alveolar opacities in < 25% of the left upper quadrant (B); chest CT demonstrated the presence of extensive parenchymal consolidation of the left lower lobe and small areas of GGO in the right lung (visual score 30%) (E). Patient 3 (male, 45 y/o) was attributed a RALE score of 36 because of the involvement of > 75% of every quadrant, with dense opacities in the left lower quadrant and moderate opacities in the remaining three quadrants (C); chest CT demonstrated bilateral consolidations (visual score 60%) (F).

measures of disease severity with the time from symptom onset. The use of these indices made it possible to better predict mortality than did the same severity measures without incorporating information about the time to reaching a certain level of disease severity. The severity/time indices based on imaging scoring of pneumonia extension were stronger predictors than those based on laboratory tests. The indices describing lung damage (based on chest imaging and sO$_2$) provided more informative value than the presence of comorbidities, which is generally considered one of the main factors driving COVID-19 mortality, along with age [2–7].

The uncovering of factors which may help to identify COVID-19 patients who will face more severe outcomes has been one of the main research goals since the beginning of the pandemic, especially for factors potentially available at ED admission. Among these factors, the most widely used are age, comorbidities, clinical and laboratory signs of lung damage, inflammation, and coagulation disorders [3–7]. The role of chest imaging has been widely evaluated, and even if the majority of available studies focus on CT scan [10–16], CXR-based scores have also been shown to be reliable in predicting COVID-19 outcomes [17–23]. This is particularly

**Table 2. Associations of pre-existing conditions and laboratory and radiological findings with death, crude and after adjustment for age and sex.**

| Variables | | Death | | | |
| --- | --- | --- | --- | --- | --- |
| | | Crude | | Multivariable | |
| | | IRR | 95% CI | IRR | 95% CI |
| Age (years) | | 1.078 | 1.053–1.103 | | |
| Sex | Female | 1 | | | |
| | Male | 2.034 | 1.092–3.789 | | |
| Charlson Comorbidity Index | 0 | 1 | | 1 | |
| | 1 | 2.549 | 1.049–6.194 | 1.231 | 0.488–3.104 |
| | 2 | 2.348 | 0.902–6.115 | 1.259 | 0.479–3.310 |
| | ≥ 3 | 4.462 | 2.446–8.137 | 1.798 | 0.947–3.412 |
| Diabetes | | 1.583 | 0.818–3.066 | 0.814 | 0.413–1.607 |
| Chronic obstructive pulmonary disease | | 4.346 | 1.966–9.604 | 1.975 | 0.878–4.443 |
| Ischemic cardiopathy | | 2.586 | 1.364–4.904 | 1.239 | 0.642–2.391 |
| Chronic kidney failure | | 3.836 | 0.935–15.743 | 1.549 | 0.373–6.428 |
| Cancer | | 1.762 | 0.961–3.232 | 1.281 | 0.698–2.350 |
| Hypertension | | 2.629 | 1.517–4.553 | 1.455 | 0.833–2.543 |
| Obesity | | 2.904 | 0.907–9.298 | 2.424 | 0.750–7.841 |
| Heart failure | | 4.496 | 2.371–8.526 | 2.682 | 1.409–5.104 |
| Arrhythmias | | 3.098 | 1.600–5.998 | 1.537 | 0.779–3.033 |
| Vascular diseases | | 2.904 | 0.907–9.298 | 2.179 | 0.680–6.983 |
| Days from symptom onset to CT | | 0.829 | 0.754–0.910 | 0.910 | 0.830–0.998 |
| CT disease extension (visual score) | | 1.036 | 1.024–1.048 | 1.028 | 1.016–1.040 |
| Days from symptom onset to CXR | | 0.835 | 0.760–0.918 | 0.915 | 0.836–1.002 |
| CXR RALE score | | 1.087 | 1.058–1.116 | 1.064 | 1.032–1.097 |
| CRP | | 1.065 | 1.032–1.098 | 1.041 | 1.007–1.076 |
| $sO_2$ | | 0.945 | 0.921–0.970 | 0.973 | 0.944–1.004 |
| Severity/time index CT (5-day lag) | | 1.334 | 1.240–1.434 | 1.223 | 1.128–1.327 |
| Severity/time index CXR (5-day lag) | | 1.737 | 1.493–2.022 | 1.411 | 1.172–1.698 |
| Severity/time index CRP (5-day lag) | | 1.944 | 1.516–2.492 | 1.419 | 1.091–1.846 |
| Severity/time index sO2 (5-day lag) | | 2.101 | 1.649–2.676 | 1.423 | 1.063–1.906 |

IRR, incidence rate ratio; CT, computed tomography; CXR, chest X-ray; RALE, Radiographic Assessment of Lung Edema; CRP, C-reactive protein; $sO_2$, oxygen saturation level. IRRs of days from symptom onset to CT and to CXR, CT visual score and CXR RALE score, CRP, sO2, and severity/time indices are for one unit increase.

important since CT is not routinely recommended by the main international guidelines unless warranted by features of respiratory worsening, especially in resource-constrained environments, where CXR is more readily available [32].

Apart from pre-existing conditions such as age, sex, and comorbidities, other predictive factors are severity measures, which provide a static picture of the disease at ED presentation. By combining these factors with the time from disease onset (time from symptom onset plus lag time), we tried to incorporate information on the evolution of the disease that led the patient to the moment of ED presentation. In order to use these indices as indicators of the velocity of disease progression, we should start from the assumption that the tested severity measures were within normal ranges at disease onset, which is not necessarily true. This limitation acknowledged, the use of these indices combining severity and time remains the only (although imperfect) way to add information on disease progression velocity before ED admission.

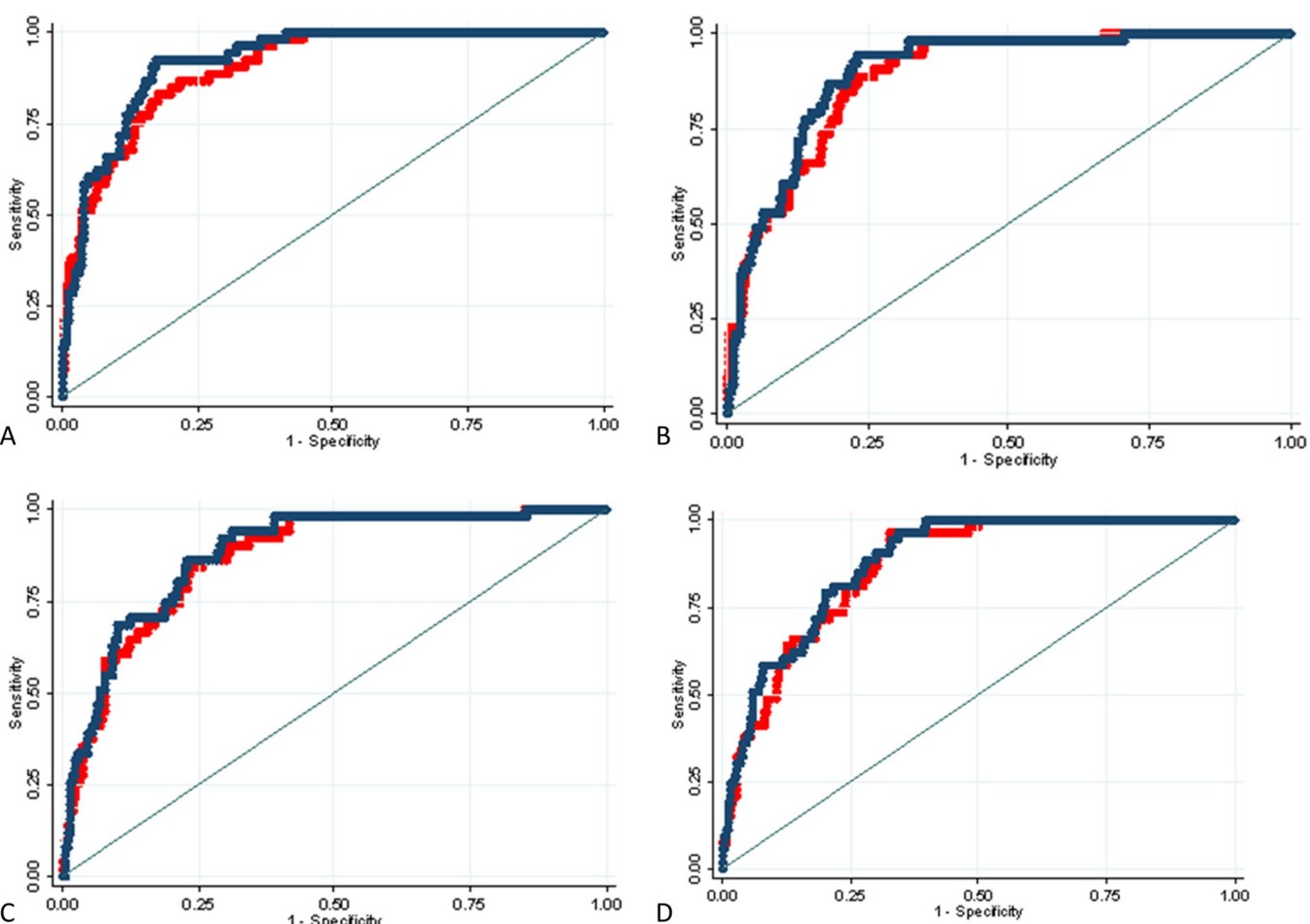

**Fig 3. ROC curves.** Receiver operating characteristic (ROC) curves of each measure of disease severity (red line) and respective severity/time index (blue line) in multivariable logistic models for death at 30 days, adjusted for age and sex. Considered measures of disease severity are computed tomography (CT) visual score of disease extension (panel A), chest X-rays (CXR) Radiographic Assessment of Lung Edema (RALE) score (panel B), CRP levels (panel C), and $sO_2$ levels (panel D). Horizontal axis: 1—Specificity from 0 to 1.00; vertical axis: Sensitivity from 0 to 1.00.

**Table 3. Crude and sex- and age-adjusted associations with death in the validation cohort.**

| Variables | | Death | | | |
|---|---|---|---|---|---|
| | | Crude | | Multivariable | |
| | | IRR | 95% CI | IRR | 95% CI |
| Age (years) | | 1.057 | 1.032–1.082 | | |
| Sex | Female | 1 | | | |
| | Male | 1.099 | 0.590–2.049 | | |
| Days from symptom onset to CXR | | 0.897 | 0.832-.968 | 0.926 | 0.861–0.997 |
| CXR RALE score | | 1.029 | 1.001–1.058 | 1.025 | 0.995–1.055 |
| Severity/time index CXR (five-day lag) | | 1.430 | 1.101–1. 857 | 1.334 | 1.014–1.756 |

IRR, incidence rate ratio; CXR, chest X-rays; RALE, Radiographic Assessment of Lung Edema; CRP, C-reactive protein; $sO_2$, oxygen saturation level. IRRs of days from symptom onset to CXR, CXR RALE score, and severity/time index are for one unit increase.

The importance of the velocity of disease progression is highlighted both by the worse outcomes of patients whose symptoms become more severe more quickly and thus present to the ED sooner after symptom onset [24–26] and by evidence that, in hospitalized patients with serial assessments, rapidly rising CT disease extension or CRP levels may predict poor outcomes [33–35]. Nonetheless, few attempts have been made to assess the velocity of disease progression before ED presentation. Information regarding the time from symptom onset to seeking medical attention was easy to collect both from the ED and in the home-care setting. This information has a strong association with all negative outcomes, such as death, hospitalization, and ED readmission [24–26]. Also, lung damage evaluated by imaging showed to be strongly associated with negative outcomes; because it is well known that signs of lung damage on imaging last even after the most severe respiratory symptoms have resolved [36], it is intuitive that placing imaging-based severity in the context of the phase of the disease progression is necessary.

In this study, including indices incorporating severity and time in the prediction model for death resulted in a better performance than the model including the measure of disease severity only. Of note, the models with the severity/time indices also slightly outperformed those including the CCI, underlining the important role of the velocity of disease progression, which is at least comparable to the much more acknowledged role of comorbidities. This was not true for CRP, however, probably because, unlike lung damage, CRP level does not increase steadily. Thus, a single measure obtained at ED admission does not permit building a reliable severity/time index.

This study has some limitations. The study collected cases retrospectively, including only those that had both CT and CXR. This condition was necessary to fulfil the study aim, but obviously introduced a selection bias since some milder cases who were not referred to CT were not included in the cohort. It is worth noting that during the study period, as the diagnostic workup for suspected COVID-19 pneumonia included CT, the vast majority of patients presenting at the ED with symptoms highly suggestive of COVID-19 underwent it [30]. In the validation cohort, all patients who underwent CXR were included, resulting in a higher prevalence of milder cases, which may explain the lower predictive value of the models in this cohort. Furthermore, as other relevant severity biomarkers, in particular D-dimer levels, were not routinely assessed at the ED during the study period, it was not possible to build another severity/time index based on the third important pathway of disease progression (i.e., coagulopathy, along with lung damage and inflammation). Finally, the impact of severity/time indices on intermediate outcomes such as hospitalization and mechanical ventilation has not been evaluated. However, while the association of indices with intermediate outcomes is plausible, it would have only led to a further increase in indices association with death.

## Conclusion

In conclusion, our study confirms that one of the most powerful prognostic factors for COVID-19 is the time from symptom onset to seeking medical assistance. This information is readily available and gives added value to the interpretation of other imaging and laboratory findings at ED presentation. Thus, our findings should encourage clinicians who evaluate a COVID-19 patient at admission to critically interpret the patient's current disease severity also in light of the time from symptom onset: the shorter the time the worse the prognosis.

## Supporting information

**S1 Table. Models for death comparing CT-, CXR-, CRP-, and sO2- based measures of disease severity and severity/time indices.**
(DOCX)

**S2 Table. Characteristics associated with death in the validation cohort.**
(DOCX)

**S1 Fig. Validation cohort ROC curves.** Receiver Operating Characteristic (ROC) curves of CXR RALE score (red line) and respective severity/time index (blue line) in multivariable logistic models for death at 30 days adjusted for age and sex. AUC values were: AUCCXR-RALE = 0.77 (95% CI, 0.71–0.84) and AUCCXR-RALE severity/time index = 0.79 (95% CI, 0.73–0.85).
(DOCX)

# Acknowledgments

We thank Jacqueline Costa for the English language editing.

# Author Contributions

**Conceptualization:** Giulia Besutti, Olivera Djuric, Marta Ottone, Guido Ligabue, Giovanni Guaraldi, Marco Massari, Francesco Luppi, Maria Giulia Galli, Pierpaolo Pattacini, Paolo Giorgi Rossi.

**Data curation:** Giulia Besutti, Olivera Djuric, Marta Ottone, Filippo Monelli, Patrizia Lazzari, Francesco Ascari, Guido Ligabue, Giuseppe Pezzuto, Petra Bechtold, Marco Massari, Ivana Lattuada, Francesco Luppi, Maria Giulia Galli, Paolo Giorgi Rossi.

**Formal analysis:** Marta Ottone, Paolo Giorgi Rossi.

**Investigation:** Giulia Besutti, Filippo Monelli, Patrizia Lazzari, Francesco Ascari, Giovanni Guaraldi, Giuseppe Pezzuto, Petra Bechtold, Ivana Lattuada, Maria Giulia Galli, Pierpaolo Pattacini, Paolo Giorgi Rossi.

**Methodology:** Giulia Besutti, Olivera Djuric, Marta Ottone, Guido Ligabue, Petra Bechtold, Francesco Luppi, Pierpaolo Pattacini, Paolo Giorgi Rossi.

**Software:** Filippo Monelli.

**Supervision:** Guido Ligabue, Giovanni Guaraldi, Marco Massari, Pierpaolo Pattacini, Paolo Giorgi Rossi.

**Validation:** Filippo Monelli, Paolo Giorgi Rossi.

**Writing – original draft:** Giulia Besutti, Olivera Djuric, Marta Ottone, Paolo Giorgi Rossi.

**Writing – review & editing:** Giulia Besutti, Olivera Djuric, Marta Ottone, Filippo Monelli, Patrizia Lazzari, Francesco Ascari, Guido Ligabue, Giovanni Guaraldi, Giuseppe Pezzuto, Marco Massari, Ivana Lattuada, Francesco Luppi, Maria Giulia Galli, Pierpaolo Pattacini, Paolo Giorgi Rossi.

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
