## [Decision Letter · Decision Letter 0]

7 Mar 2022

PONE-D-21-29668Imaging-based Indices of the Velocity of Disease Progression to Predict COVID-19 Mortality: A Cohort StudyPLOS ONE

Dear Dr. Besutti,

Thank you for submitting your manuscript to PLOS ONE. After careful consideration, we feel that it has merit but does not fully meet PLOS ONE’s publication criteria as it currently stands. Therefore, we invite you to submit a revised version of the manuscript that addresses the points raised during the review process.

We look forward to receiving your revised manuscript.

Kind regards,

Antonino Salvatore Rubino, M.D., Ph.D.

Academic Editor

PLOS ONE

Reviewers' comments:

Reviewer's Responses to Questions

**Comments to the Author**

1. Is the manuscript technically sound, and do the data support the conclusions?

Reviewer #1: Yes

Reviewer #2: Yes

2. Has the statistical analysis been performed appropriately and rigorously? 

Reviewer #1: I Don't Know

Reviewer #2: Yes

3. Have the authors made all data underlying the findings in their manuscript fully available?

Reviewer #1: No

Reviewer #2: No

4. Is the manuscript presented in an intelligible fashion and written in standard English?

Reviewer #1: Yes

Reviewer #2: Yes

5. Review Comments to the Author

Reviewer #1: PONE-D-21-29669 Reviewer comment

I would like to commend the authors for the time and dedication towards this manuscript. Technically, this is a good work however, I have some issues with the study design which need to be addressed.

1. Aim of the study versus what was done

Aim / Purpose of the study was stated as: “The study aim was to evaluate indices of the velocity of disease progression to predict mortality of COVID-19 patients admitted to the emergency department (ED).”

Your title and various sections in your manuscript referred to “velocity of disease progression”.

The term “velocity of disease progression” conferred the notion that, there were at least teo measurements of same severity measure within a time interval and then the velocity of progression computed from the change in severity measure over the time.

However, in your described methods, the time interval was measured from time of onset of symptoms to time of presentation at the emergency department, when the various tests were carried out. This measured interval directly measures, time from onset of symptoms to time of presentation at the emergency department which is influenced by various determinants including several factors contributing to delays in decision to seek care, delays in arriving at the facility and delays in the hospital before the measured tests were carried out. Please check this and revise you’re the aim and title of the study appropriately.

2. Velocity indexes

Also linked with the above comment, is a concern about the computation of the velocity index as described in lines 141- 147. “In order to obtain indices that could indicate the velocity of disease progression before the first ED admission, we combined the CT visual score, the CXR RALE score, CRP levels, and sO2 levels separately with the time elapsed from symptom onset and the measurement of these parameters. The easiest way to obtain a velocity was to divide the severity measure by the time needed to reach that level of severity. This strategy was used for CRP levels and for CT- and CXR-based extension of parenchymal involvement, while sO2 velocity index was calculated as the difference in sO2 value to 100 (100-sO2) divided by the time from symptom onset to sO2 measurement.”

This approach assumes that the baseline estimate for each severity measure was normal range and same value for all study participants, so that “dividing the severity measure by the time needed to reach that level of severity” would give a composite measure that is comparable among all study participants. However, based on past medical history of each study participants, the severity measure ie. sO2, CXR RALE, CT visual score, CRP at baseline which was not measured and practically difficult to determine based on the current study methodology, may be abnormal. This approach at estimating velocity index did not account for these variations in estimate at baseline. Please critically consider this.

3. Interventions carried out

There was no mention of the interventions at the emergency department which might have prevented outcome of death or delayed it beyond the 30days mark. I feel this is an important omission.

4. Retrospective study or prospective study?

Study design and selection of participants section, line 86 stated; “This prospective cohort study included all consecutive patients aged > 18 years who presented to…..”

And in same section, line 90 -91 stated that “Given 91 the retrospective nature of the study, the Ethics Committee authorized the use of a patient’s data….”

Please revise and be consistent

5. Conclusion of the study

Conclusion in abstract “Indices describing the velocity of COVID-19 progression, especially those based on imaging, better predicted mortality than the same severity measures not incorporating the time needed to reach a certain level of severity.

Conclusion as stated in the last paragraph of the discussion stated “In conclusion, our study confirms that one of the most powerful prognostic factors for COVID-19 is the time from symptom onset to seeking medical assistance. This information is readily available and gives added value to the interpretation of other imaging and laboratory findings at ED presentation.”

The conclusion in the abstract need to be revised to conform with what was stated in the last paragraph of the discussion

6. Minor concerns

a) Introduction, lines 48-49, “…more than 140 million cases and 3 million deaths have been reported globally up to May 2021 [1]” I will suggest a more current statistic.

b) Discussion section, lines 263-267 stated that “The role of chest imaging has been widely evaluated, and even if the majority of available studies focus on CT scan [10-16], CXR-based scores have also been shown to be reliable in predicting COVID-19 outcomes [17-23]. This is particularly important since CT is not routinely recommended by the main international guidelines unless warranted by features of respiratory worsening, especially in resource-constrained environments, where CXR is more readily available [31]”

This is true and the implication on your method of selection such that only patients with CT scan and Chest X-ray were included suggested that this study primarily admitted patients with features of worsening respiratory features. This limitation needs to be acknowledged.

Thank you

Reviewer #2: Appreciating your work, please find hereunder some comments and suggestions:

1. Can the authors explicitly define their inclusion and exclusion criteria? Or provide an explanation as to why they forwent on an exclusion criteria?

2. Can the authors provide their ethical statement in a separate segment?

3. "Given the retrospective nature of the study, the Ethics Committee authorized the use of a patient’s data without his/ her informed consent if all reasonable efforts had been made to contact that patient." Can the authors provide a number or a citation that states/further explains this clause?

4. Can the authors write their conclusion in a separate segment and add some statements that highlight the implication of their findings to clinical and global/public health practice?

6. PLOS authors have the option to publish the peer review history of their article (what does this mean?). If published, this will include your full peer review and any attached files.

Reviewer #1: No

Reviewer #2: No

---

## [Author Response · Author response to Decision Letter 0]

29 Apr 2022

PONE-D-21-29669 Reviewer comment

Reviewer #1: I would like to commend the authors for the time and dedication towards this manuscript. Technically, this is a good work however, I have some issues with the study design which need to be addressed.

Re: We thank the Reviewer for the positive appraisal and for having underlined some important issues, allowing us to improve our work.

1. Aim of the study versus what was done

Aim / Purpose of the study was stated as: “The study aim was to evaluate indices of the velocity of disease progression to predict mortality of COVID-19 patients admitted to the emergency department (ED).”

Your title and various sections in your manuscript referred to “velocity of disease progression”.

The term “velocity of disease progression” conferred the notion that, there were at least teo measurements of same severity measure within a time interval and then the velocity of progression computed from the change in severity measure over the time. 

However, in your described methods, the time interval was measured from time of onset of symptoms to time of presentation at the emergency department, when the various tests were carried out. This measured interval directly measures, time from onset of symptoms to time of presentation at the emergency department which is influenced by various determinants including several factors contributing to delays in decision to seek care, delays in arriving at the facility and delays in the hospital before the measured tests were carried out. Please check this and revise you’re the aim and title of the study appropriately.

Re: We thank the Reviewer for this comment. We agree on the confounding factors (delays in decision to seek care/in arriving at hospital/in performing tests) which can influence the time between symptom onset and disease severity measure. However, even if it is an imperfect indicator, time between symptom onset and emergency department admission has been shown to be one of the most important predictive factors of poor outcome (see references 24-26). On the other hand, we usually use different laboratory and imaging severity measures as prognostic/predictive factors even if they represent a static picture of the patient at the time of admission (so, even more imperfect). We thought that the only possible way to include both pieces of information (time and severity) in a single parameter was to compute these indices, to have an indication on the speed at which the disease is worsening. However, we agree with the Reviewer that since we do not have two timepoints for the measurement of disease severity measures, speaking of “velocity” is incorrect: we changed the definition “velocity index” to “severity/time index”. We used the severity measure instead of a variation between two timepoints starting from the assumption that each severity measure was normal before disease onset. We agree that this assumption must be explicit when we discuss the opportunity to use these indices as measures of disease progression velocity. Nevertheless, when a patient arrives at the ED, we only have this information to estimate prognosis. 

We changed the title and the aim into “Imaging-based Indices Combining Disease Severity and Time from Disease Onset to Predict COVID-19 Mortality: A Cohort Study” and “The aim of this study was to evaluate the role of indices that combine radiological data at admission and time from disease onset, in predicting mortality of COVID-19 patients admitted to the ED.” respectively. Throughout the text we changed “velocity indices” to “severity/time indices”. 

We have changed a paragraph in the discussion in order to add the aforementioned assumption:

“Apart from pre-existing conditions such as age, sex, and comorbidities, other predictive factors are severity measures, which provide a static picture of the disease at ED presentation. By combining these factors with the time from disease onset (time from symptom onset plus lag time), we tried to incorporate information on the evolution of the disease that led the patient to the moment of ED presentation. In order to use these indices as indicators of the velocity of disease progression, we should start from the assumption that the tested severity measures were within normal ranges at disease onset, which is not necessarily true. This limitation acknowledged, the use of these indices combining severity and time remains the only (although imperfect) way to add information on the velocity of disease progression before ED admission.”

2. Velocity indexes

Also linked with the above comment, is a concern about the computation of the velocity index as described in lines 141- 147. “In order to obtain indices that could indicate the velocity of disease progression before the first ED admission, we combined the CT visual score, the CXR RALE score, CRP levels, and sO2 levels separately with the time elapsed from symptom onset and the measurement of these parameters. The easiest way to obtain a velocity was to divide the severity measure by the time needed to reach that level of severity. This strategy was used for CRP levels and for CT- and CXR-based extension of parenchymal involvement, while sO2 velocity index was calculated as the difference in sO2 value to 100 (100-sO2) divided by the time from symptom onset to sO2 measurement.”

This approach assumes that the baseline estimate for each severity measure was normal range and same value for all study participants, so that “dividing the severity measure by the time needed to reach that level of severity” would give a composite measure that is comparable among all study participants. However, based on past medical history of each study participants, the severity measure ie. sO2, CXR RALE, CT visual score, CRP at baseline which was not measured and practically difficult to determine based on the current study methodology, may be abnormal. This approach at estimating velocity index did not account for these variations in estimate at baseline. Please critically consider this.

Re: We thank the Reviewer for underlining this point. We have added a paragraph on this in the discussion (see the answer above).

3. Interventions carried out

There was no mention of the interventions at the emergency department which might have prevented outcome of death or delayed it beyond the 30days mark. I feel this is an important omission.

Re: We should clarify that the indices were not used as criteria to decide whether to hospitalize or not COVID-19 patients, in fact these indices have been calculated retrospectively (we have now clarified the study design, see following answer). Still, indices are obviously associated with disease severity measures that were used for patient stratification. All severe COVID-19 patients were treated in the same way during the first pandemic wave (hospitalization and oxygen therapy, and if necessary mechanical ventilation, tocilizumab and/or steroids, which were only administered to hospitalized patients at the time). It is plausible that indices are associated with intermediate outcomes such as hospitalization, but should this be the case, this bias should go in a conservative direction with more intensive therapies given to the cases with worst prognostic indices, thus therapies could partially improve the observed prognosis. In the methods section we have added a short paragraph on the therapeutic protocol during the first pandemic wave in Reggio Emilia, while in the limitation section we have added a paragraph explaining that the lack of evaluation of the impact of severity/time indices on intermediate outcomes such as hospitalization may have resulted in an under- but not over-estimation of associations of indices with death.

Methods: “During the first pandemic wave in Reggio Emilia, patients presenting to the ED with fever and SpO2>95% were discharged home in case of negative chest X-rays and/or CT or in case of positive chest X-rays and/or CT scan but who were >70 years of age and had no relevant past medical history. Patients >70 years of age and/or with relevant past medical history could be hospitalized even without respiratory failure, while others were admitted to hospital in case of radiological findings of pneumonia combined with respiratory failure. In the case of radiological findings of complicated pneumonia with or without acute respiratory distress syndrome (ARDS), patients started non-invasive ventilation in the ED and were admitted to subintensive/intensive care unit.” We added a reference for this point.

Limitations: “Finally, the impact of severity/time indices on intermediate outcomes such as hospitalization and mechanical ventilation has not been evaluated. However, while the association of indices with intermediate outcomes is plausible, it would have only led to a further increase in indices association with death.”

4. Retrospective study or prospective study?

Study design and selection of participants section, line 86 stated; “This prospective cohort study included all consecutive patients aged > 18 years who presented to…..”

And in same section, line 90 -91 stated that “Given 91 the retrospective nature of the study, the Ethics Committee authorized the use of a patient’s data….”

Re: We thank the Reviewer and we understand that the classification is misleading. We have removed the term prospective, as the cohort definition was retrospective. However, we have added a clarification on the fact that information that has been registered at the time of ED presentation (e.g., laboratory data or CT disease extension) could not have been modified at the time of retrospective data collection: “Even if the researchers assessed inclusion criteria and conditions at admission retrospectively, all the information registered at the time of ED presentation (e.g. laboratory data or CT disease extension) was not modifiable later, therefore exposures could not be influenced by the occurrence of the outcome.”

5. Conclusion of the study

Conclusion in abstract “Indices describing the velocity of COVID-19 progression, especially those based on imaging, better predicted mortality than the same severity measures not incorporating the time needed to reach a certain level of severity.

Conclusion as stated in the last paragraph of the discussion stated “In conclusion, our study confirms that one of the most powerful prognostic factors for COVID-19 is the time from symptom onset to seeking medical assistance. This information is readily available and gives added value to the interpretation of other imaging and laboratory findings at ED presentation.”

The conclusion in the abstract need to be revised to conform with what was stated in the last paragraph of the discussion

Re: We thank the Reviewer for the suggestion. The conclusion in the abstract has been changed to “Time from symptom onset to ED admission is a strong prognostic factor and provides added value to the interpretation of imaging and laboratory findings at ED presentation.”

6. Minor concerns

a) Introduction, lines 48-49, “…more than 140 million cases and 3 million deaths have been reported globally up to May 2021 [1]” I will suggest a more current statistic.

Re: Data have been updated to March 2022.

b) Discussion section, lines 263-267 stated that “The role of chest imaging has been widely evaluated, and even if the majority of available studies focus on CT scan [10-16], CXR-based scores have also been shown to be reliable in predicting COVID-19 outcomes [17-23]. This is particularly important since CT is not routinely recommended by the main international guidelines unless warranted by features of respiratory worsening, especially in resource-constrained environments, where CXR is more readily available [31]”

This is true and the implication on your method of selection such that only patients with CT scan and Chest X-ray were included suggested that this study primarily admitted patients with features of worsening respiratory features. This limitation needs to be acknowledged.

Re: We thank the Reviewer for the comment. In the limitation section, we have already stated: “The study collected cases retrospectively, including only those that had both CT and CXR. This condition was necessary to fulfil the study aim, but obviously introduced a selection bias since some milder cases who were not referred to CT were not included in the cohort. It is worth noting that during the study period, as the diagnostic workup for suspected COVID-19 pneumonia included CT, the vast majority of patients presenting at the ED with symptoms highly suggestive of COVID-19 underwent it [30].” In order to better explain the point, we have added a sentence in the study design section, explaining that during the first COVID-19 wave in Reggio Emilia (which was months before the publications of guidelines that discouraged the routinary use of CT), CT was performed by almost all patients presenting to the ED for suspected COVID-19: “During the COVID-19 outbreak, the diagnostic protocol for suspected COVID-19 patients presenting to EDs included RT-PCR, blood tests, chest X-rays, and CT in every case of suggestive X-rays or negative X-rays but with highly suggestive clinical features.”

Reviewer #2: Appreciating your work, please find hereunder some comments and suggestions:

1. Can the authors explicitly define their inclusion and exclusion criteria? Or provide an explanation as to why they forwent on an exclusion criteria?

Re: We thank the Reviewer for the comment. We rephrased the sentence at the beginning of the study design as follows: “This retrospective cohort study included all consecutive patients meeting the following inclusion criteria: patients > 18 years of age; presenting to any one of the provincial EDs between 27 February and 13 March 2020; and with a positive RT-PCR within 10 days from ED admission. Patients who did not undergo both CT scan and CXR at ED presentation were excluded.

2. Can the authors provide their ethical statement in a separate segment?

Re: A segment named “Ethical approval” has now been added.

3. "Given the retrospective nature of the study, the Ethics Committee authorized the use of a patient’s data without his/ her informed consent if all reasonable efforts had been made to contact that patient." Can the authors provide a number or a citation that states/further explains this clause?

Re: This sentence is the translation of what is reported by the Ethics Committee in its authorization to conduct the study, according to Italian law in terms of privacy (GDPR n. 679/2016, D.Lgs. 196/2003, modified by D.Lgs. 101/2018 and Measure by the Personal Data Protection Guarantor n. 146 date 05/06/2019)

4. Can the authors write their conclusion in a separate segment and add some statements that highlight the implication of their findings to clinical and global/public health practice?

Re: We thank the Reviewer for the suggestion. A segment named “Conclusion” has been created, and a sentence on implications has been added: “Thus, our findings should encourage clinicians who evaluate a COVID-19 patient at admission to critically interpret the patient’s current disease severity also in light of the time fro

---

## [Decision Letter · Decision Letter 1]

6 Jun 2022

Imaging-based Indices Combining Disease Severity and Time from Disease Onset to Predict COVID-19 Mortality: A Cohort Study

PONE-D-21-29668R1

Dear Dr. Besutti,

We’re pleased to inform you that your manuscript has been judged scientifically suitable for publication and will be formally accepted for publication once it meets all outstanding technical requirements.

Kind regards,

Antonino Salvatore Rubino, M.D., Ph.D.

Academic Editor

PLOS ONE

Additional Editor Comments (optional):

The Reviewers positively accepted Your comments

Reviewers' comments:

Reviewer's Responses to Questions

**Comments to the Author**

1. If the authors have adequately addressed your comments raised in a previous round of review and you feel that this manuscript is now acceptable for publication, you may indicate that here to bypass the “Comments to the Author” section, enter your conflict of interest statement in the “Confidential to Editor” section, and submit your "Accept" recommendation.

Reviewer #1: (No Response)

Reviewer #2: All comments have been addressed

2. Is the manuscript technically sound, and do the data support the conclusions?

Reviewer #1: Yes

Reviewer #2: Yes

3. Has the statistical analysis been performed appropriately and rigorously? 

Reviewer #1: Yes

Reviewer #2: Yes

4. Have the authors made all data underlying the findings in their manuscript fully available?

Reviewer #1: No

Reviewer #2: No

5. Is the manuscript presented in an intelligible fashion and written in standard English?

Reviewer #1: Yes

Reviewer #2: Yes

6. Review Comments to the Author

Reviewer #1: Thank you for the opportunity to review this manuscript. The authors have taken the necessary steps to fully address all my concerns

Reviewer #2: All questions and comments have been addressed. No further feedback or suggestions. I thank the authors for making the time to integrate these comments into their manuscript to better the scientific content.

7. PLOS authors have the option to publish the peer review history of their article (what does this mean?). If published, this will include your full peer review and any attached files.

Reviewer #1: No

Reviewer #2: No

---

## [Editor Report · Acceptance letter]

9 Jun 2022

PONE-D-21-29668R1 

Imaging-based Indices Combining Disease Severity and Time from Disease Onset to Predict COVID-19 Mortality: A Cohort Study 

Dear Dr. Besutti:

I'm pleased to inform you that your manuscript has been deemed suitable for publication in PLOS ONE. Congratulations! Your manuscript is now with our production department. 

Kind regards, 

on behalf of

Dr. Antonino Salvatore Rubino 

Academic Editor

PLOS ONE